# Differential Effects of Extracellular Vesicles of Lineage-Specific Human Pluripotent Stem Cells on the Cellular Behaviors of Isogenic Cortical Spheroids

**DOI:** 10.3390/cells8090993

**Published:** 2019-08-28

**Authors:** Mark Marzano, Julie Bejoy, Mujeeb R. Cheerathodi, Li Sun, Sara B. York, Jing Zhao, Takahisa Kanekiyo, Guojun Bu, David G. Meckes, Yan Li

**Affiliations:** 1Department of Chemical and Biomedical Engineering, FAMU-FSU College of Engineering, Florida State University, Tallahassee, FL 32310, USA; 2Department of Biomedical Sciences, College of Medicine, Florida State University, Tallahassee, FL 32306, USA; 3Department of Neuroscience, Alzheimer’s Disease Research Center, Mayo Clinic, Jacksonville, FL 32225, USA

**Keywords:** induced pluripotent stem cells, extracellular vesicles, neural progenitors, neural degeneration, cardiac mesoderm

## Abstract

Extracellular vesicles (EVs) contribute to a variety of signaling processes and the overall physiological and pathological states of stem cells and tissues. Human induced pluripotent stem cells (hiPSCs) have unique characteristics that can mimic embryonic tissue development. There is growing interest in the use of EVs derived from hiPSCs as therapeutics, biomarkers, and drug delivery vehicles. However, little is known about the characteristics of EVs secreted by hiPSCs and paracrine signaling during tissue morphogenesis and lineage specification. Methods: In this study, the physical and biological properties of EVs isolated from hiPSC-derived neural progenitors (ectoderm), hiPSC-derived cardiac cells (mesoderm), and the undifferentiated hiPSCs (healthy iPSK3 and Alzheimer’s-associated SY-UBH lines) were analyzed. Results: Nanoparticle tracking analysis and electron microscopy results indicate that hiPSC-derived EVs have an average size of 100–250 nm. Immunoblot analyses confirmed the enrichment of exosomal markers Alix, CD63, TSG101, and Hsc70 in the purified EV preparations. MicroRNAs including miR-133, miR-155, miR-221, and miR-34a were differently expressed in the EVs isolated from distinct hiPSC lineages. Treatment of cortical spheroids with hiPSC-EVs in vitro resulted in enhanced cell proliferation (indicated by BrdU+ cells) and axonal growth (indicated by β-tubulin III staining). Furthermore, hiPSC-derived EVs exhibited neural protective abilities in Aβ42 oligomer-treated cultures, enhancing cell viability and reducing oxidative stress. Our results demonstrate that the paracrine signaling provided by tissue context-dependent EVs derived from hiPSCs elicit distinct responses to impact the physiological state of cortical spheroids. Overall, this study advances our understanding of cell‒cell communication in the stem cell microenvironment and provides possible therapeutic options for treating neural degeneration.

## 1. Introduction

Extracellular vesicles (EVs), including exosomes and microvesicles, are membrane-bound cellular products (30–1000 nm) that contain various classes of nucleic acids as well as soluble and transmembrane proteins [1]. EVs are described based on size, sub-cellular origin, proposed functions, and biogenesis and release pathways [2]. EVs play important roles in intercellular communications, immune modulation, senescence, proliferation, differentiation, and maintaining tissue homeostasis [3,4,5,6]. In addition, EVs can cross the blood‒brain barrier through two proposed mechanisms: (1) internalization by endothelial cells, transcytosis, and release on the other side of the barrier; or (2) entry into the central nervous system through the intercellular junctions of endothelial cells [7]. Together, these properties make EVs attractive targets for the studies of disease mechanisms and the development of drug delivery systems.

Stem-cell-derived EVs have major applications as therapeutic biologics (cell-free) for in vivo delivery to promote endogenous progenitor proliferation, stimulate angiogenesis, remodel extracellular matrices, and regulate immune responses [8,9]. They can be further modified to deliver therapeutic small molecules, proteins, and RNAs [10]. For example, EVs can functionally transfer proteins, small interfering RNA (siRNA), mRNAs, and microRNAs to the target cells [11,12]. EVs are also intensely studied for diagnostic applications in cancer, neurology, and immunology [13,14]. For example, EVs have been collected from bodily fluids (e.g., blood) and tested for the expression of G protein-coupled receptors (GPCRs) on chips for GPCR agonists or antagonists for early diagnostics [6].

There is growing interest in the characterization of dendritic cell-derived EVs and mesenchymal stem cell (MSC)-derived EVs for various applications. However, few studies have characterized human induced pluripotent stem cell (hiPSC)-derived EVs. Human iPSC-derived MSCs (iMSCs) exhibit increased cell survival and improved proliferation and differentiation when compared with adult MSCs [15,16]. As EVs have been found to participate in these cellular processes, differences in the contents of iMSC-EVs may account for the improved cell survival and proliferation. In particular, hiPSCs have shown great potential in generating physiologically relevant neural cells [17], tissues, and brain spheroids or organoids [18,19,20,21,22,23,24,25,26,27,28,29] for studying neurological disease progression and neurotropic viral infections (e.g., Zika virus) [30,31,32,33,34]. Some disease pathologies (e.g., amyloid-β plaques) may take years to develop, making them difficult to study in vivo. However, the EVs produced by cultured brain organoids can be used to probe the early stages of disease onset for the identification of biomarkers [24,35]. Recently, the fusion of brain organoids of dorsal or ventral regions has provided a feasible approach to study interneuron migration [36,37,38]. The assembly of vascular spheroids, mesenchymal cells, and cortical spheroids was developed by our group to study heterotypic cell-to-cell interactions [39,40]. However, the paracrine signaling elicited by EVs from different cell types in the human brain and during neural tissue morphogenesis has not been well investigated.

Based on the available data, we hypothesize that the properties of hiPSC-EVs are determined by the developmental stage, lineage specification, and genetic backgrounds of the stem cells. For instance, culture passage number and cell density have been found to impact the yield and vascularization activities of MSC-derived EVs [41,42]. Previous studies suggest that undifferentiated hiPSC-EVs likely play a major role in maintaining cells in a pluripotent state and providing cellular protective effects [43,44,45]. In addition, the EVs of iPSC-derived mesoderm cells (cardiomyocytes or MSCs) have been shown to elicit anti-apoptotic responses in recipient cells [46,47]. Moreover, EVs offer an intriguing new avenue of research with regard to Alzheimer’s disease (AD) because they have recently been found to mediate cellular communications and to transport protein and microRNA cargos specifically associated with AD pathology [7]. Despite the importance of EVs in AD pathogenesis, the global alterations in EV cargo and subsequent contributions to the disease remain unknown. Therefore, hiPSCs provide a broad platform to create developmental and lineage-specific populations for the production and characterization of EVs to better understand brain tissue morphogenesis and disease progression due to paracrine signaling. Consequently, hiPSC-EVs potentially provide not only new avenues for biomarker discovery, but also novel therapeutics for AD and other neurodegenerative diseases [48,49].

To test our hypothesis, EVs were harvested and characterized from healthy and AD-associated hiPSCs, which were maintained as undifferentiated, or differentiated into mesoderm or neural progenitors. The EVs were examined for size distribution, the presence of exosomal markers, and the expression of some specific microRNAs that regulate Wnt signaling and exhibit anti-apoptotic activity. The influences of iPSC-EVs on the viability and proliferation of cortical spheroids were further investigated. Moreover, the neuroprotective ability of different hiPSC-EVs on cortical spheroids stimulated with neurotoxic amyloid β (Aβ) 42 oligomers was evaluated.

## 2. Materials and Methods

### 2.1. Undifferentiated hiPSC Culture

Human iPSK3 cells were derived from human foreskin fibroblasts transfected with plasmid DNA encoding reprogramming factors OCT4, NANOG, SOX2 and LIN28 (kindly provided by Dr. Stephen Duncan, Medical College of Wisconsin, and Dr. David Gilbert, Department of Biological Sciences of Florida State University) [50,51]. Human iPSK3 cells were maintained in mTeSR serum-free medium (StemCell Technologies, Inc., Vancouver, Canada) on six-well plates coated with growth factor reduced Geltrex (Life Technologies, Carlsbad, CA, USA). The cells were passaged by Accutase dissociation every 5–6 days and seeded at 1 × 106 cells per well of tissue culture-treated six-well plate (in 3 mL medium) in the presence of 10 µM Y27632 (Sigma, St. Louis, MO, USA) for the first 24 h [52,53,54]. The conditioned media from day 4–7 of the culture were collected 24 h after the previous medium change.

Patient-specific hiPSC with presenilin-1 (PS1) mutation, the SY-UBH cell line, was derived from the fibroblasts of an early-onset AD individual with PS1 M146V mutation (kindly provided by Dr. Takahisa Kanekiyo and Dr. Guojun Bu, Alzheimer’s Disease Research Center, Mayo Clinic, Jacksonville, FL, USA). The SY-UBH cells were maintained in mTeSR serum-free medium on Matrigel (growth factor reduced, BD Biosciences, San Jose, CA, USA)-coated surface [55]. The cells were passaged by dispase and mechanical scraping every seven days at a 1:3 split ratio in the presence of Y27632 (10 µM) for the first 24 h. The conditioned media from day 4–7 of the culture were collected 24 h after the previous medium change.

### 2.2. Cardiovascular Differentiation from iPSK3 Cells

Human iPSK3 cells were seeded on 24-well plates coated with Geltrex (Life Technologies) at a cell density of 3 × 105 cells/well in 1 mL culture medium. Cells were maintained in mTeSR serum-free medium with 10 µM Rho kinase inhibitor Y27632 (Sigma) for the first day and in mTeSR medium for another four days. The cardiomyocyte differentiation was induced by modulating Wnt pathways with small molecules CHIR99021 (a Wnt activator) and IWP4 (a Wnt inhibitor) (Giwi protocol) [56]. Briefly, cells were cultivated in Roswell Park Memorial Institute (RPMI) medium plus 2% B27 serum-free supplement minus insulin medium (Life Technologies) with 5–10 µM CHIR99021 (day 0) for 24 h and then CHIR99021 (StemCell Technologies, Inc.) was withdrawn from the medium. After another two days, the cells were then cultivated in the medium with 5 µM IWP4 (Stemgent, Cambridge, MA, USA) for two days. At day 6 or 7, the medium was changed to RRMI plus 2% B27 and beating cells were observed at day 12–14 [57]. The conditioned media from day 15–20 culture were collected 48 h after the previous medium change.

### 2.3. Cortical Spheroid Differentiation of iPSK3 Cells

Human iPSK3 cells were seeded into Ultra-Low Attachment 24-well plates (Corning Incorporated, Corning, NY, USA) at 3 × 105 cells/well in 1 mL of differentiation medium composed of Dulbecco’s Modified Eagle Medium/Nutrient Mixture F-12 (DMEM/F-12) plus 2% B27 serum-free supplement (Life Technologies). Y27632 (10 µM) was added during the seeding and removed after 24 h. At day 1, the cells formed embryoid bodies (EBs) and were treated with dual SMAD signaling inhibitors 10 µM SB431542 (Sigma) and 100 nM LDN193189 (Sigma) [53,54,58]. After eight days, the cells were treated with fibroblast growth factor (FGF)-2 (10 ng/mL, Life Technologies) and retinoic acid (RA) (5 µM, Sigma) until day 16. At day 17 the spheroids were replated onto Geltrex-coated cell plates. The conditioned media from day 15–20 culture were collected 48 h after the previous medium change.

### 2.4. Total EV/Exosome Isolation

For exosome isolation, about 8–10 mL of conditioned medium from each group (pooled from the referred collecting days) were centrifuged at 2000 *g* for 30 min to remove cells and debris. The cell-free supernatants were filtered through a 0.22-µm membrane and transferred to a new tube. The filtered supernatants were concentrated about 20 times using a 100-kDa filter (Amicon Ultra15, Millipore) and then incubated with a 0.5 volume of Total Exosome Isolation Reagent (Thermo Fisher, Waltham, MA, USA) [59]. The mixture was incubated at 2–8 °C overnight. The supernatant/reaction mixture was centrifuged at 10,000 *g* for 1 h at 2–8 °C. The supernatants were discarded and the EV/exosome-containing pellets were collected for subsequent experiments.

Alternatively, the differential ultracentrifugation method was used to isolate iPSC-exosomes for characterization by Western blot. The conditioned media were centrifuged at 500 *g* for 5 min at 4 °C. The supernatants were removed and centrifuged again at 2000 *g* for 10 min. The supernatants were removed again and centrifuged at 10,000 *g* for 30 min. The ultracentrifugation was performed for supernatants at 100,000 *g* for 70 min. The supernatants were discarded and the pellets were resuspended with phosphate buffer saline (PBS) and centrifuged at 100,000 *g* for 70 min. The EV/exosome-containing pellets were collected for subsequent experiments. In addition, a polyethylene glycol (PEG)-based method was used to isolate the EVs as reported previously [49].

### 2.5. Protein Assay

Protein content was measured by the Bradford assay (Thermo Fisher), using bovine serum albumin (BSA) as a standard. Specifically, 5 µL of exosome preparation was added to 250 µL Coomassie reagent, and incubated for 10 min at room temperature. The protein concentration was quantified by measuring the absorbance at 595 nm on a microplate reader.

### 2.6. Immunocytochemistry

Briefly, the samples were fixed with 4% paraformaldehyde (PFA) and permeabilized with 0.2–0.5% Triton X-100. The samples were then blocked for 30 min and incubated with various mouse or rabbit primary antibodies (Appendix A) for 4 h. After washing, the cells were incubated with the corresponding secondary antibodies for 1 h. The samples were counterstained with Hoechst 33342 and visualized using a fluorescent microscope (Olympus IX70, Melville, NY, USA). For 5-Bromo-2′-deoxyuridine (BrdU) assay, the day 25 cortical spheroid outgrowth was incubated in medium containing 10 µM BrdU (Sigma) for 8 h. The samples were then washed and fixed with 70% cold ethanol for 30 min at 2–8 °C. A denaturation step was then performed using 2N HCl/0.5% Triton X-100 incubation at room temperature for 30 min in the dark. The samples were neutralized with 1 mg/mL sodium borohydride for 5 min. After washing, the samples were incubated with mouse anti-BrdU (1:100, Life Technologies) in a blocking buffer (0.5% Tween 20/1% BSA in PBS) for 30–60 min at room temperature, followed by Alexa Fluor® 488 goat anti-Mouse IgG1 incubation for 30 min. The cells were counterstained with Hoechst 33342 and analyzed by a fluorescent microscope. Using ImageJ software (http://rsb.info.nih.gov/ij), the percentages of BrdU+ cells were calculated as the ratios of green intensity (i.e., surface area covered by green signals) over nuclear intensity (provided by Hoechst 33342 staining).

### 2.7. Flow Cytometry Analysis

Since the forward scatter and side scatter parameters can be used to detect EVs/exosomes [60], the iPSC-derived EVs/exosomes were isolated and analyzed by flow cytometry. The samples were acquired with BD FACSCanto™ II flow cytometer (Becton Dickinson, Franklin Lakes, NJ, USA) and analyzed against PBS control using FlowJo software (FlowJo, LLC, Ashland, Oregon, USA). For LIVE/DEAD assay, the live cells were stained for 1 µM calcein-AM (green) and 2 µM ethidium homodimer I (red) for 30 min. The samples were acquired along with the compensation controls. For immunophenotyping, about 1 × 10^6^ cells per sample were fixed with 4% PFA and washed with staining buffer (2% fetal bovine serum in PBS). The cells were permeabilized with 100% cold methanol, blocked, and then incubated with various primary antibodies (Appendix A) followed by the corresponding secondary antibody. The cells were acquired and analyzed against isotype controls using FlowJo software.

### 2.8. Nanoparticle Tracking Analysis

Nanoparticle tracking analysis (NTA) was performed on fresh exosome samples in triplicate using a Nanosight LM10-HS (Malvern Instruments, Malvern, UK) configured with a blue (488 nm) laser and sCMOS camera to estimate size distribution and particle concentration [49]. Samples were diluted to 1–2 µg/mL in PBS, and three videos of 60 s were acquired for each replicate. The camera shutter speed was fixed at 30 ms. NTA3.0 software (Malvern, UK) was used to measure the mode size, i.e., light-scatter (standard) mode, the mean size, and the concentration of particles per 1 mL solution. The mode size is the vesicle size that appears most often and is usually a more accurate representation of the EV size as vesicle aggregates affect the mean size. The camera level was set to 13, and the detection threshold was maintained at 3 to ensure accurate and consistent detection of small particles. The laser chamber was cleaned thoroughly between each sample reading with particle-free water and 70% ethanol.

### 2.9. Transmission Electron Microscopy

Following EV isolation, exosome-enriched isolates were resuspended in 50–100 µL of sterile filtered PBS for electron microscopy imaging according to Lasser et al. [61]. Briefly, intact exosomes (5 µL) were dropped onto Parafilm for each sample preparation. With forceps, a carbon-coated 400 Hex Mesh Copper grid (Electron Microscopy Sciences, Hatfield, PA, USA) was positioned with coating side down on top of each drop for 60 min. Grids were then washed with sterile filtered PBS three times, and the grid was transferred on top of a 30 µL drop of PBS each time. Excess solution was wicked off the grid gently using absorbing paper between each wash, while avoiding direct contact with the coating side. Exosomes were fixed by positioning the grid on top of a 20-µL drop of 2% PFA (EM grade, Electron Microscopy Sciences) for 10 min. After washing, grids were then transferred on top of a 20 µL drop of 2.5% glutaraldehyde (EM Grade, Electron Microscopy Sciences) for 10 min, followed by three washes with particle-free filtered water. Grid samples were stained on a 20 µL drop of 2% uranyl acetate (EMS grade) for 10 min before embedding on 20 µL of 0.13% methyl cellulose and 0.4% uranyl acetate for another 10 min. Excess liquid was removed using absorbing paper and grids were left to dry coated side up before imaging on the electron microscope.

### 2.10. iPSC-Derived Neural Progenitor (iNPC) Spheroids Treated with Aβ (1–42) Oligomers and EVs

To prepare oligomers of the Aβ42 peptide, biotinylated Aβ42 (Bachem, Bubendorf, Switzerland) was fully dissolved at 0.5 mg/mL in hexafluor-2-propanole (HFIP, Sigma) [54,62]. HFIP Aβ(1–42) solution was dispensed at 10 µL into each siliconized Snap-Cap microtube. The microtubes were put in a desiccator to completely evaporate HFIP, and thereafter stored at −80 °C. Oligomer solutions were prepared fresh for each experiment. The stock was dissolved in 10 µL of DMSO (to 105 µM) and incubated for 3 h at room temperature. Oligomers of Aβ42 were added to the day 20–30 cortical neural cultures (in 96-well plates) derived from human iPSK3 cells at 1 µM, based on our previous publications [54,62]. EVs from undifferentiated iPSK3, SY-UBH, iPSK3-mesoderm (cardiovascular cells), and iPSK3-ectoderm (NPC) were added at 50 µg protein/mL at the same time. LIVE/DEAD assay and reactive oxygen species assay were performed for Aβ42-oligomer-treated cultures. BrdU assay and neural axon characterization (β-tubulin III staining) were performed for the exosome-treated culture (day 25).

### 2.11. Live/Dead Assay

The cells were evaluated for viability using a Live/Dead® staining kit (Molecular Probes, Eugene, OR, USA). After 72 h of Aβ42 oligomer and iPSC-EV treatments, the cells were washed and incubated in DMEM-F12 containing 1 µM calcein-AM (green) and 2 µM ethidium homodimer I (red) for 30 min. The samples were imaged under a fluorescent microscope (Olympus IX70). Using ImageJ software, the viability was analyzed and calculated as the ratio of green intensity (i.e., surface area covered by green signals) over total intensity (surface area covered by all cells, including both green cells and red cells) or by flow cytometry.

### 2.12. Reactive Oxygen Species (ROS) Assay

ROS detection was performed using Image-iT™ Live Green Reactive Oxygen Species Detection kit (molecular probes) [63,64]. Briefly, the spheroids and single cells were washed in Hank’s Balanced Salt Solution and incubated in a solution of 25 µM carbioxy-H2DCFDA for 30 min at 37 °C. The samples (with or without Aβ42 oligomers stimulation) were then washed and analyzed under a fluorescence microscope or by flow cytometry. Using ImageJ software, the ROS expression ratio was calculated as the ratio of green intensity (i.e., surface area covered by green signals) over nuclear intensity (provided by Hoechst 33342 staining). As a positive control, the cells were incubated in a 100 µM tert-butyl hydroperoxide solution, prior to staining with carbioxy-H2DCFDA.

### 2.13. Western Blotting for Exosomal Markers

EV pellets (with equal number of particle counts) were resuspended in 0.2 µm filtered PBS (based on NTA results) and lysed by adding an equal volume of a urea-containing strong lysis buffer. The EV samples were then boiled for 5 min, centrifuged, and an equal volume of lysate (35 µL) was resolved on a 10% running/4% stacking SDS-PAGE gel for 45 min at 100 V until the loading dye entered the resolving layer, then at 150 V until the end. Proteins were transferred at 4 °C onto nitrocellulose membranes. Then the blots were blocked with 5% milk in PBS-Tween 20 solution for 30 min. After blocking, the membranes were probed with CD63, TSG101, Alix, and Hsc70 (all 1:1000 dilution, Appendix A) in 2.5% BSA in PBS-Tween 20 solution overnight at 4 °C. After washing, each blot was incubated with anti-rabbit or anti-mouse secondary antibody (1:3000 dilution) for 1 h at room temperature. Finally, blots were washed, a chemiluminescent substrate (Azure Biosystems, Dublin, CA, USA) was added, and then they were imaged using IMAGEQUANT LAS400 (GE Healthcare Life Sciences, Pittsburgh, PA, USA).

### 2.14. MicroRNA Reverse Transcription Polymerase Chain Reaction (RT-PCR)

Total microRNA (miRNA) was isolated from different iPSC-EV samples using the miRNeasy Micro Kit (Qiagen, Valencia, CA, USA) according to the manufacturer’s protocol. Reverse transcription was carried out using commercial qScript miR cDNA synthesis kit (Quantabio, Beverly, MA, USA). The PerfeCTa® Universal PCR Primer (QuantaBio) has been designed and validated to work specifically with PerfeCTa SYBR Green SuperMix using miRNA cDNA produced. The levels of miR-133, miR-155, miR-221, and miR-34a were determined. *U6, SNORD48,* and *SNORD44* were examined as housekeeping genes for the normalization of miR expression levels (primer sequences are shown in Table 1). Real-time RT-PCR reactions were performed on an Applied Biosystems Quantstudio 7 flex (Applied Biosystems, Foster City, CA, USA), using SYBR1 Green PCR Master Mix (Applied Biosystems). The amplification reactions were performed as follows: 10 min at 95 °C, and 40 cycles of 95 °C for 15 s and 60 °C for 30 s, and 70 °C for 30 s. Fold variation in gene expression was quantified by means of the comparative Ct method: 2^(−(∆C_(t treatment)−∆C_(t control)), which is based on the comparison of expression of the target gene (normalized to the endogenous control) between the compared samples.

### 2.15. Statistical Analysis

The representative experiments were presented and the results were expressed as (mean ± standard deviation). To assess the statistical significance, one-way ANOVA followed by Fisher’s LSD post hoc tests were performed for multiple groups and the comparisons between two groups were analyzed by *t*-tests. A *p*-value < 0.05 was considered statistically significant.

## 3. Results

### 3.1. Isolation and Characterization of hiPSC-EVs

To study EVs from different developmental lineages, iPSK3 cells were first differentiated into cardiomyocytes (mesoderm) and neural progenitors (ectoderm) (Figure 1). At the undifferentiated stage, the cells exhibited the typical morphology of hPSCs with a high nuclei to cytoplasm ratio and colony-like morphology (Figure 1A). Performing a LIVE/EAD assay on the iPSK3 cell culture revealed a cell viability of 76.0% (Appendix A). Following cardiovascular differentiation, the cells expressed cardiac marker α-actinin and Nkx2.5 (Figure 1B and Appendix A). The differentiation efficiency, as measured by α-actinin, ranged from 23.7% to 84.8%, as cardiac differentiation from human PSCs need strict culture conditions and a narrow time window for small molecule treatment [57]. For neural progenitor differentiation, the cells were allowed to form spheroids and then replated to visualize the outgrowth. Axonal growth was readily visible (Figure 1C), and the cells expressed neural progenitor marker Nestin (>70%) [54] and neuron marker β-tubulin III (>40%) (Appendix A) [54]. The neuronal cells contain both glutamatergic and GABAergic neurons based on our previous publication [53]. Condition media were collected from various cultures for downstream EV characterization.

The EVs isolated from the culture media were analyzed for protein levels (Figure 2A). The EV protein amounts were 2.3–2.4 µg protein per mL of spent medium from all of the three groups. The particle concentration (by NTA analysis) was also similar between EVs from the different producer cells. NTA analysis revealed that the average particle size of EVs isolated from the three groups differed. EVs from iPSK3-Mesoderm (Meso) and iPSK3-neural progenitors (Ecto) showed an average size of 164.6 ± 5.3 nm and 182.6 ± 7.2 nm, respectively (Figure 2A). The average size of EVs from undifferentiated iPSK3 cells (241.8 ± 8.4 nm) was greater than that in the other two groups. The mode size was also bigger for undifferentiated iPSK3 (125.6 ± 4.9 nm) than for the iPSK3-Meso group (99.3 ± 1.5 nm), but comparable to that of the iPSK3-Ecto group (119.1 ± 9.4 nm). The mode size for NTA is usually a better representation of the size as large protein or EV aggregates can significantly affect the mean size. Similar NTA results were observed using a PEG-based isolation method (Appendix A).

EVs from iPSK3 and SY-UBH undifferentiated cells were also compared (Figure 2B). The protein content was 2.3–2.4 µg protein per mL of spent medium. NTA analysis showed average mean sizes at 185. 5 ± 11.1 nm (114.1 ± 15.2 nm for mode size) and 165.5 ± 4.2 nm (112.9 ± 2.4 nm for mode size) respectively. The particle concentration (by NTA analysis) was similar for the two groups.

The endosomal sorting complexes required for transport (ESCRT) is highly involved in exosome production and protein trafficking [1,65]. Alix and TSG101 are part of ESCRT machinery and CD63 is a tetraspanin commonly found on the surface of exosomes. Undifferentiated iPSK3 cells expressed abundant levels of CD63 and TSG101 (Figure 3A(i)). Heat shock cognate 71 kDa protein (Hsc70), also known as HSPA8, is a member of the heat shock protein 70 family (Hsp70) and commonly found in EVs [65]. The levels of these exosomal markers in EVs purified from neural-ectoderm (iPSK3-Ecto), cardiac-mesoderm (iPSK3-Meso), and undifferentiated iPSK3, as well as the SY-UBH group, were confirmed by Western blot (Figure 3A(ii) and Appendix A). The smeary appearance of the CD63 band in Figure 3A(ii)a is due to the presence of glycosylated CD63 in varying degrees. The reduced glycosylation is shown in Figure 3A(ii)b. CD63 was lower in the iPSK3-Ecto group than the iPSK3-Meso and the iPSK3 undifferentiated groups. Alix, TSG101, and Hsc70 were highly expressed in the iPSK3-Meso group, suggesting that the iPSC-derived NPCs may utilize different protein machinery for biogenesis or trafficking. The SY-UBH group had undetectable Alix and TSG101, and the lowest expression of Hsc70. The absence of Calnexin expression, a negative marker of EVs, was observed in the derived EVs (but present in cell lysate) (Appendix A). The total protein stain was performed using Ponceau S to confirm the equal protein loading for Figure 3A(ii)a (Appendix A).

A flow cytometry method was used to detect the EVs of different groups [60]. While the forward scatter (FSC) and side scatter (SSC) plots were distinct from the PBS control, showing the presence of nanoscale particles, it was difficult to determine the EV size (Appendix A). The accurate flow cytometry analysis of EVs may require the use of fluorescent dyes or markers and sensitive flow cytometers like the BD influx instrument [66]. To further confirm the presence of EVs, preparations were analyzed by electron microscopy. Small round particles with typical cup-shaped morphology were identified in all samples, verifying that EVs were harvested (Figure 3B) [67].

### 3.2. Effects of hiPSC-EVs on the Proliferation and Axonal Growth of iNPC Spheroids

We next wanted to test the biological effects of the EVs on spheroid cultures. To do this, hiPSC-EVs from the different cell lineages were added to the day 25 iNPC spheroids and the spheroids were replated onto a Geltrex-coated surface. After 3 d, the cell proliferation was examined using BrdU incorporation, which indicates the cells in S-phase of cell cycle (Figure 4). The iPSK3-Ecto and iPSK3-Meso EVs resulted in comparable BrdU+ cells as the control. However, EVs from the undifferentiated iPSK3 cells increased the percentage of BrdU+ cells (68.4 ± 3.0% vs. 49.± 6.6%). In contrast, the SY-UBH group showed fewer BrdU+ cells (30.4 ± 4.2% vs. 49.9 ± 6.6%) compared to the control. The percentages of BrdU+ cells were generally high, probably due to the long incubation time of the cells with BrdU during incorporation.

Similarly, iNPC cultures were treated with EVs derived from the three groups and the axonal length (indicated by β-tubulin III staining) was analyzed (Figure 5) [68]. The iNPCs treated with iPSK3-Ecto EVs had the greatest axon length (423.5 ± 36.7 µm), compared to the control group (348.0 ± 25.9 µm), while the undifferentiated iPSK3 and iPSK3-Meso EVs did not promote axonal growth compared to the control. This may be due to the fact that iPSK3-Ecto EVs contain cargo that is specifically beneficial for neural cells, including maintaining the integrity and growth of the axons.

### 3.3. Effects of hiPSC-EVs on Aβ42 Oligomer-Treated iNPC Spheroids

Aβ(1–42) is a major component of the neuritic plaques found in Alzheimer’s disease and can induce apoptosis and cell death in cultured neurons [54,55,62]. To test the protective abilities of iPSC-derived EVs, iNPC spheroids were treated with Aβ42 oligomers and the cultures with or without EVs from the iPSC groups were subjected to a LIVE/DEAD assay (Figure 6 and Appendix A). Cells treated with Aβ42 oligomers only and SY-UBH-derived EVs had the lowest level of viability (0.82 ± 0.11) compared to the other four groups (0.94–0.98). The three iPSK3-EV groups did not show significant differences in viability compared to the culture without Aβ42 oligomers. These data suggest that iPSC-EVs from healthy cell sources have neural protective effects.

ROS assay can test for the amount of oxidative stress, which is related to cell death and particularly relevant for neurodegenerative diseases such as AD. In these experiments, iNPCs treated with Aβ42 oligomers served as a positive control, in which high oxidative stress was observed (Figure 7A). The treatment of Aβ42 oligomer-stimulated cultures with different EV groups reduced the levels of ROS. The reduction, based on image analysis, was the greatest for undifferentiated iPSK3 group, followed by the iPSK3-Ecto, iPSK3-Meso, and SY-UBH groups (Figure 7B). In addition to image analysis, flow cytometry was performed to quantify the reduction in ROS levels compared to the level of iNPCs treated with Aβ42 oligomers (Figure 7C). The ROS reduction was consistently more pronounced with EVs from the undifferentiated iPSK3, iPSK3-Ecto, and iPSK3-Meso groups, but less so for the SY-UBH group.

### 3.4. Differential Expression of miRNAs in Different iPSC-EV Groups

As an initial assessment of the molecular effects of different iPSC-EV groups on cellular behaviors, miRNAs were isolated and analyzed from the three types of iPSC-EVs. The qPCR for the total miRNAs identified the groups with the greatest and least abundance of miRNAs as the iPSK3-UD and iPSK3-Meso groups, respectively (Appendix A). Several house-keeping genes were tested including U6, SNORD44, and SNORD48 (Appendix A). For our samples, U6 had more variations than SNORD44 among different samples (not among replicates). For SNORD48, one sample had completely different melting curves compared to the other two samples. Therefore, SNORD44 was selected as the best housekeeping gene for normalization purposes (Figure 8i,ii). miR-133-3p, miR-133b, and miR-155-5p were highly expressed in the iPSK3-Ecto group (0.94–1.12), and least in the iPSK3-UD group (0.14–0.18) (Table 1 and Figure 8ii). For miR-155-3p, the iPSK3-Meso group had the highest level of expression (1.93 ± 0.10), followed by the iPSK3-Ecto (0.90 ± 0.14) and iPSK3-UD groups (0.25 ± 0.31). For miR-221-3p, the iPSK3-Meso group also had the highest level of expression (1.31 ± 0.04), followed by the iPSK3-UD group (1.18 ± 0.06) and the iPSK3-Ecto group (1.00 ± 0.00), while no significant difference was observed for miR-221-5p (Figure 8iii). For miR-34a-3p, it was highly expressed in the iPSK3-Ecto group (1.21 ± 0.29) and less so in the iPSK3-Meso and iPSK3-UD groups (0.33–0.39). However, for miR-34a-5p, the iPSK3-Meso group had the greatest level of expression (6.86 ± 0.04), followed by the iPSK3-UD (3.76 ± 0.31) and iPSK3-Ecto groups (0.94 ± 0.09) (Figure 8iv). These data indicate the differential expression of distinct microRNA cargo in lineage-specific iPSC-derived EVs, which may contribute to the differential cellular responses or represent distinguishing biomarkers.

## 4. Discussion

iPSC-EVs, enriched with proteins and miRNA, may be used for identifying biomarkers during disease progression, as vehicles for drug delivery, or as therapeutics. It is likely that iPSC-EVs provide a better therapeutic platform compared to iPSC-based therapy. For example, iPSC-EVs are safer (no tumor formation) than parent iPSCs for cardiac repair in vivo [69]. However, before the full potential of iPSC-EVs can be realized, a better understanding of their molecular cargo and biogenesis mechanisms must be obtained. Unlike MSC-derived EVs, the characterization of iPSC-EVs has not been well described in the literature.

In this study we provide an initial characterization of EVs produced by different iPSC groups. Our culture and isolation methods generate about 2–3 µg of EV protein per mL of spent medium, which is in the EV yield range of 1–10 µg/mL of culture supernatants described in other studies using similar methods [70]. No significant difference was found for the yields of EVs of different iPSC groups. Culture methods and purification approaches can affect EV yields [41]. For example, a hollow fiber bioreactor system was developed for scalable production of EVs associated with heterodimeric interleukin-15 [70]. Compared to a conventional T-flask, a hollow fiber culture system produces about 4-fold more EVs. In addition to culturing methods, the EV isolation method can influence the EV yield. For example, PEG-based precipitation methods result in greater EV yields compared to ultracentrifugation [49]. In our study, the derived iPSC-EVs (except the SY-UBH group) all expressed the typical exosomal markers CD63, TSG101, Alix, and Hsc70. However, the total exosome isolation (a polymer precipitation method based on PEG) and PEG methods of EV enrichment also co-purify protein and RNA complexes. To obtain highly purified EVs for proteomics or RNA-Seq, this method should be combined with Optipep density gradient purification [49,71]. It will also be interesting to see if more highly purified EV preparations possess greater neuroprotective properties when compared to equivalent amounts of PEG-enriched EV samples.

For EVs derived from undifferentiated hiPSCs, our results show that the cells receiving the iPSK3-EVs have the highest proliferation capacity (indicated by BrdU+ cells). For cells treated with Aβ42 oligomers, the EVs reduced the oxidative stress significantly and improved the cell viability, but had no effect on axonal growth. Undifferentiated iPSCs were reported to release about 2200 EVs/cell/h in the first 12 h with an average size of 122 nm [45], producing 16-fold more EVs than various types of MSCs in a chemically defined medium [72]. mRNAs in iPSC-EVs may contain reprogramming factors like Oct3/4, Nanog, Klf4 and c-Myc. Glycome analysis of EVs using high-density lectin microarray showed that the characteristic glycan signature of hiPSCs was captured in hiPSC-EVs [43]. hiPSC-EVs were also shown to restore cell viability and capillary-like structure formation, and reduce senescence in HUVECs exposed to high glucose [73]. Furthermore, hiPSC-EVs have been reported to stimulate the proliferation and migration of human dermal fibroblasts [74]. In particular, hiPSC-EVs reduce ROS levels of senescent MSCs, improve the growth of replicatively aged MSCs, and alleviate cellular aging in a genetically induced senescent model. These effects are thought to be due in part to delivering intracellular peroxiredoxin antioxidant enzymes (e.g., PRDX1 and PRDX2) [72]. In addition, hiPSC-EVs can be used as natural gene delivery vectors to reduce the inflammatory response of recipient cells [75]. Taken together, hiPSC-EVs have enormous potential for various therapeutic applications.

For EVs derived from hiPSC-differentiated cardiovascular cells (iCM, representing mesoderm), our results reveal that the cells receiving the iCM-EVs exhibit similar proliferation (indicated by BrdU+ cells) to the control and the iCM-EVs have no effect on axonal growth. For the cells treated with Aβ42 oligomers, the iCM-EVs reduced oxidative stress (not as much as iPSK3-EVs) and improved the cell viability. iPSCs reprogrammed from cardiac fibroblasts generated EVs that can protect H9C2 cells from H2O2-induced oxidative stress through caspase 3/7 inhibition [76]. Moreover, iCM-EVs, enriched in heat shock proteins (HSP20, 27, 60, 70, 90), exert protective effects and alter the transcriptome and proteomic profiles of the recipient cells [77]. In addition to proteins, EVs also contain many different miRNA species that have been shown to possess important functions in cell-to-cell communications. Therefore, the characterization of miRNAs in different hiPSC derivative groups (as seen in Figure 8) will be important. Indeed, iCM-EVs are enriched with miRNAs and 16 of them are involved in tissue repair and angiogenesis [77,78]. Cardioprotective miRs or cardiac-specific miRNAs were also found in iCM-EVs—for example, miR-1, 21, and 30 [9,59].

For EVs derived from hiPSC-differentiated neural progenitor cells (iNPCs, representing ectoderm), our results indicate that cells receiving the EVs exhibit similar proliferation (indicated by BrdU+ cells) to the control, but promoted axonal growth. For cells treated with Aβ42 oligomers, EVs were observed to reduce oxidative stress and improve cell viability. The activation of the extracellular signal-regulated kinase pathway and the release of growth factors (e.g., EGF) by NPCs may contribute to the observed effects [79]. It was reported that EVs from hiPSC-derived neurons contained more mid-region tau than full-length tau [2], which can move from cell to cell through EVs. Therefore, exosomal tau in the EVs of hiPSC-NPCs might be used as a biomarker for AD. EVs derived from hPSC-neural stem cells (NSCs) have also been used for stroke treatment [8,80]. In a comparison with iPSC-MSC EVs, iPSC-NSC EVs promoted macrophage polarization toward an anti-inflammatory phenotype and increased the regulatory T-cell population in vivo after thromboembolic stroke. The iPSC-NSC EVs also reduced lesion volume (based on T2-weighted sequences) and improved behavioral outcomes (e.g., coordination on a balance beam) in aged mice. The mechanisms include anti-oxidative, pro-angiogenic, immunomodulatory, and neural plasticity regulating processes. The direct comparison of EVs from iNPCs and primary human brain NPCs still remains to be done.

Our results further demonstrate that SY-UBH cell-derived EVs reduce cell proliferation and have a limited ability to reduce oxidative stress. However, they were unable to improve the viability of cells treated with Aβ42 oligomers. We observed lower expression of Alix and TSG101 in the EVs from SY-UBH cells compared to the other groups, suggesting reduced exosome production compared to other extracellular vesicle subpopulations secreted from this group. Alternatively, an alternative mechanism of exosome biogenesis (e.g., ceramide pathway) may be predominantly utilized by SY-UBH cells. Similarly, the apolipoprotein E4 genotype was reported to result in lower EV levels and reduced TSG101 in the extracellular space within the brain, likely due to a downregulation of EV biogenesis and secretion from the endosomal pathway [81]. Therefore, the mutation in PS1 M146V of SY-UBH cells may also compromise exosome production.

It is clear that the EV cargo can reflect the pathological status of iPSC lines. For example, iPSCs derived from aged donor cells (A-iPSCs) and from young donor cells (Y-iPSCs) were compared for their secretion of RNA‒exosome complexes [44]. A-iPSCs-secreted exosomes, which have poor expression of ZSCAN10, can elevate glutathione peroxidase 2 (i.e., GPX2), leading to excess glutathione-mediated ROS scavenging activity (imbalance of ROS and glutathione homeostasis). EVs may play competing roles in neural degeneration [82]. On the one hand, EVs can modulate the phagocytic clearing of misfolded protein such as Aβ peptides. On the other hand, EVs can promote the extracellular release of toxic proteins such as tau, SOD1, TOP-43, and prions. Therefore, the characterization of iPSC-derived EVs can provide insight into the extracellular microenvironment and paracrine signaling that contribute to disease pathogenesis [6].

Previous research establishes miRNAs as having essential roles in intercellular communications via EVs [1,11]. RNA profiling for hPSC-derived cortical neurons and oligodendrocytes demonstrates the significance of miRNAs in neural cell function [83,84]. Our analysis detected several important miRNAs in iPSC-EVs, including miR-133, which may be involved in cell proliferation and neurite outgrowth [85,86]. For example, miR-133a suppresses the expression of apoptotic proteins caspase-3, 8, 9, and improves the expression of Bcl-2. The higher miR-133a and miR-133b in the ectoderm group may contribute to better axonal growth (Figure 5). In addition, miRNAs regulate a majority of Wnt signaling components, which are critical regulators of development and disease [87,88]. miR-155 targets β-catenin interacting proteins and miR-221 targets the transcription factor HDAC6 in the canonical Wnt pathway (Appendix A). Since the cardiovascular differentiation procedure involves Wnt activation using CHIR99021, the expression of miR-155-3p and miR-221-3p was observed to be higher in the mesoderm group. However, the higher miR-155-3p and miR-221-3p may not be the only contributors to cell proliferation and cell viability. miR-34a regulates the PD-1/PD-L1 pathway and is indirectly related to the p53 pathway for modulating cell death [89]. The 5′ and 3′ strands represent 5p and 3p miRNAs, respectively [90]. The differential expression of miR-133, miR-155, miR-221, and miR-34a in different iPSC-EV groups indicates the lineage-specific nature of miRNA cargo. Taken together, our results demonstrate the feasibility of analyzing miRNA cargo and performing large-scale miRNA profiling of iPSC-EVs.

## 5. Conclusions

Extracellular vesicles and the cargo they carry are cell-specific and reflect the physiological state of the producer cells. In this study, NTA results show that there is no significant difference in the amount of hiPSC-derived EVs with different lineage specifications. However, the derived EVs exhibit differential neural protective ability and the capability to reduce oxidative stress of cortical spheroids. Neural-specific EVs appear to better maintain healthy and longer axons. Differences in miRNA (miR-133, 155, 221, 34a) cargo were observed in EVs from different groups, which may account for the distinct biological properties observed. This study indicates that the paracrine signaling provided by tissue context-dependent EVs derived from hiPSCs elicits distinct signaling to impact the physiological state of cortical spheroids.

## Figures and Tables

**Figure 1 cells-08-00993-f001:**
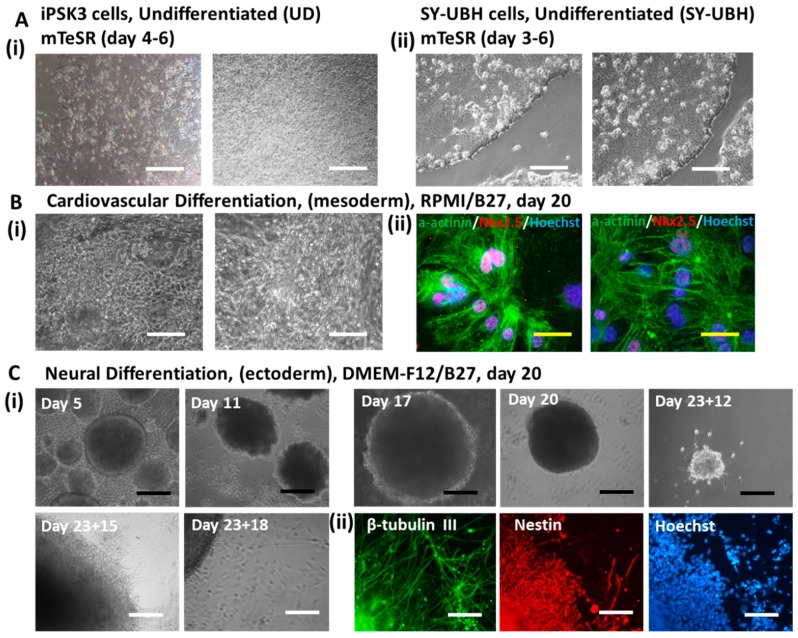
Illustration of different hiPSC cultures for EV/exosome isolation. (**A**) Undifferentiated (i) iPSK3 cells (healthy) and (ii) SY-UBH cells (from an unidentified early-onset Alzheimer’s patient); (**B**) cardiovascular mesoderm differentiation from iPSK3 cells: (i) culture morphology at day 20; (ii) expression of α-actinin and Nkx2.5 for cardiomyocyte identity. The two pictures show the variations in morphology of different cultures during the days that the conditioned media were collected. (**C**) Generation and characterization of ectodermal neural progenitor cells (NPCs) from iPSK3 cells: (i) cortical spheroid morphology over time. The cortical spheroids were replated at day 23 on Geltrex-coated surface. The axons grew out of the spheroids. (ii) Expression of Nestin and β-tubulin III for neuron identity (day 24). Scale bar: white—100 µm, yellow—25 µm, black—200 µm.

**Figure 2 cells-08-00993-f002:**
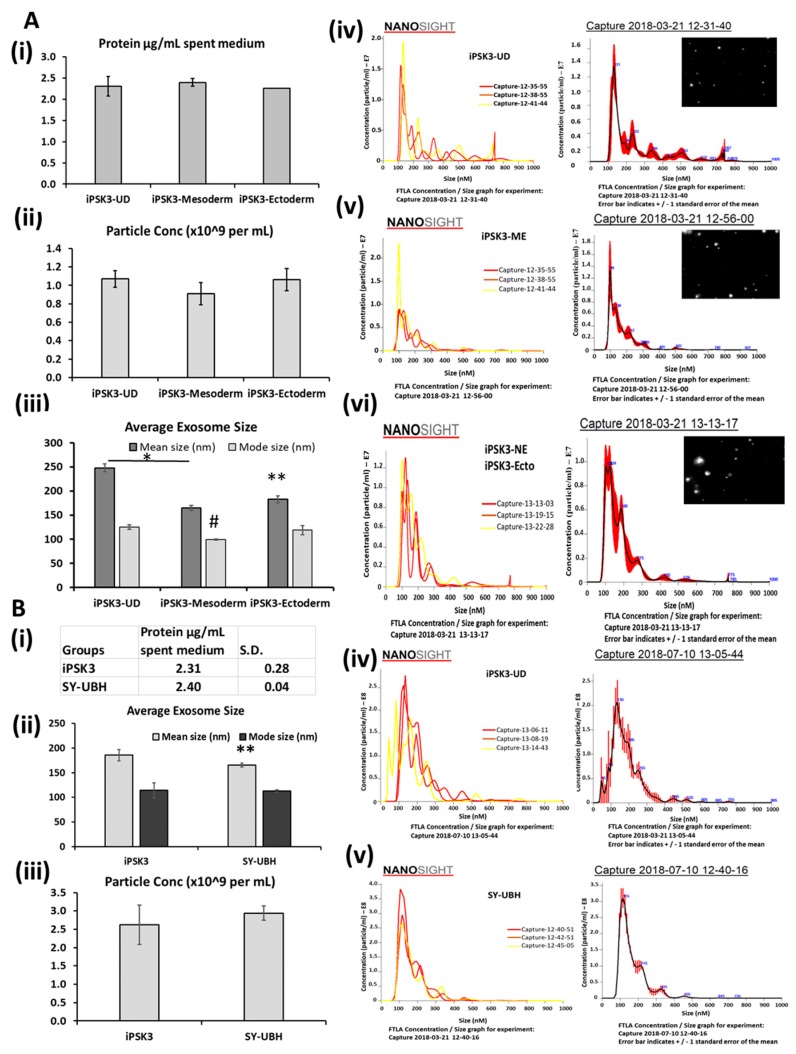
Characterization of hiPSC-derived EVs by nanoparticle tracking analysis (NTA). (**A**) Comparison of EVs from undifferentiated, mesoderm, and ectoderm (i.e., NPC) cultures derived from iPSK3 cells: (i) protein content per mL spent medium; the error for the iPSK3-ectoderm group is very small/zero; (ii) particle concentration. Particles were in 1 mL of particle-free phosphate buffer saline (PBS), which originated from 12 mL of spent media; (iii) mean and mode average particle size. Representative NTA histogram for (iv) undifferentiated; (v) mesoderm; (vi) ectoderm. The insets show the representative light scattering images. (**B**) Comparison of EVs from undifferentiated iPSK3 cells and SY-UBH cells: (i) protein content per mL spent medium; (ii) particle concentration. Particles were in 1 mL of particle-free PBS, which originated from 22 mL of spent media. (iii) Mean and mode average particle size. Representative NTA histogram for (iv) iPSK3 cells and (v) SY-UBH cells. * and ** indicate *p* < 0.05 for mean size, and # indicates *p* < 0.05 for the mode size between the compared conditions and the undifferentiated iPSK3 group (*n* = 3).

**Figure 3 cells-08-00993-f003:**
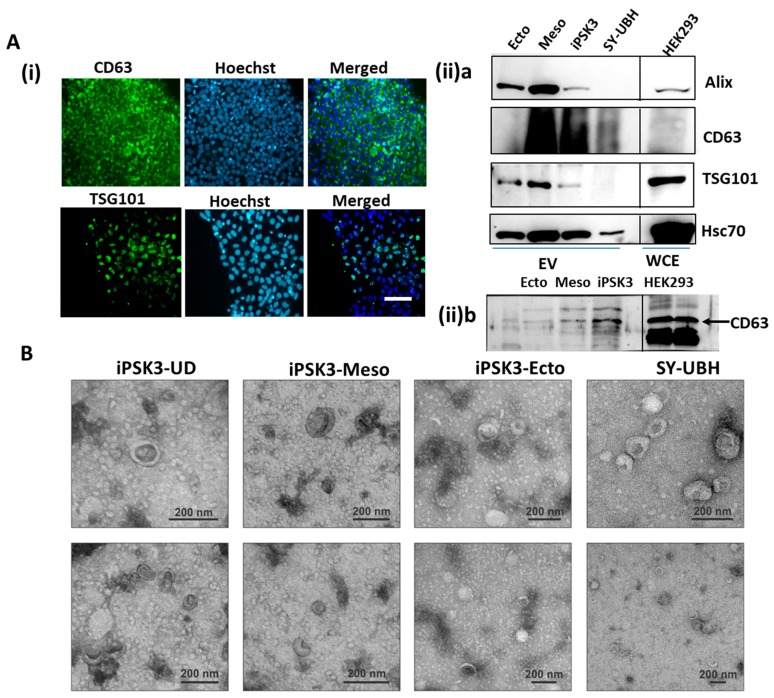
Characterization of hiPSC-derived EVs for exosomal markers and morphology. (**A**) (i) CD63 and TSG101 expression on undifferentiated iPSK3 cells; Scale bar: 100 µm. (ii) a: Western blot analysis of Alix, CD63, TSG101, and Hsc70 for the derived EVs from different iPSK3 groups as well as the SY-UBH group. WCE: whole cell extracts. WCE from HEK293 cells was used as a positive control. b: Western blot analysis of CD63 for the derived EVs from different iPSK3 groups, showing distinct CD63 bands. (**B**) Presence of exosome-sized, cup-shaped vesicles was verified by electron microscopy (EM). Scale bar: 200 nm.

**Figure 4 cells-08-00993-f004:**
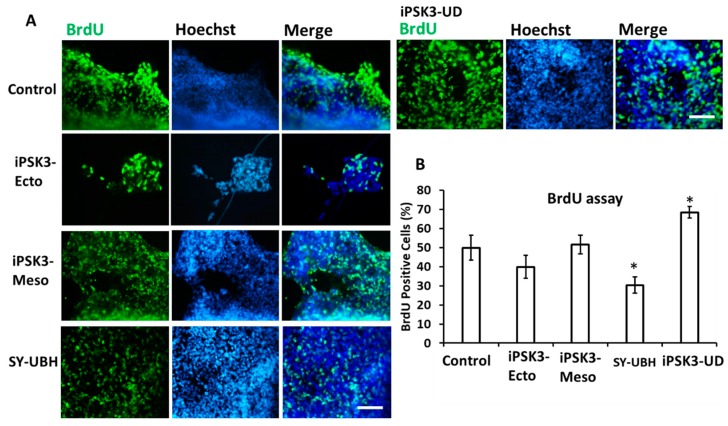
Effects of hiPSC-derived EVs on cell proliferation of iNPC spheroid outgrowth. Day 25 cortical spheroid outgrowth was treated with different groups of hiPSC-EVs for three days before the analysis. Cell proliferation was measured using BrdU assay (detect cells in S-phase of the cell cycle). (**A**) Fluorescent images of BrdU positive cells; Blue: Hoechst. (**B**) Quantification of BrdU positive cells (*n* = 3). Scale bar: 100 µm. * indicates *p* < 0.05 compared to the control condition. The control condition is the cultures treated without exosomes.

**Figure 5 cells-08-00993-f005:**
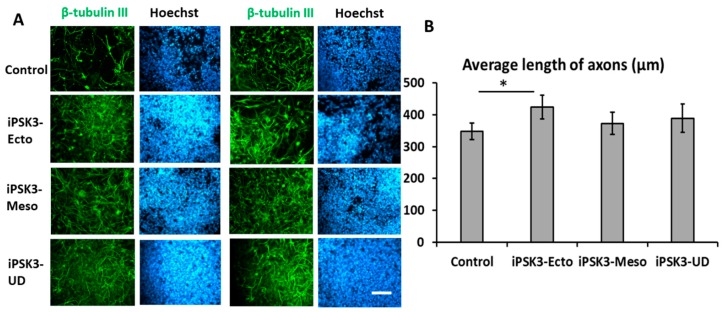
Effects of hiPSC-derived EVs on axonal growth of iNPC spheroid outgrowth by β-tubulin III staining. Day 25 cortical spheroid outgrowth was treated with different groups of hiPSC-EVs for three days before the analysis. The length of each axon was quantified by ImageJ software. Twenty axon lengths were averaged for each image. Three images were analyzed for each group. The data from the three images were averaged to give the overall average axon length. (**A**) Fluorescent images of β-tubulin III positive cells; Blue: Hoechst. (**B**) Quantification of axon length indicated by β-tubulin III expression. Scale bar: 100 µm. * indicates *p* < 0.05 compared to the control condition.

**Figure 6 cells-08-00993-f006:**
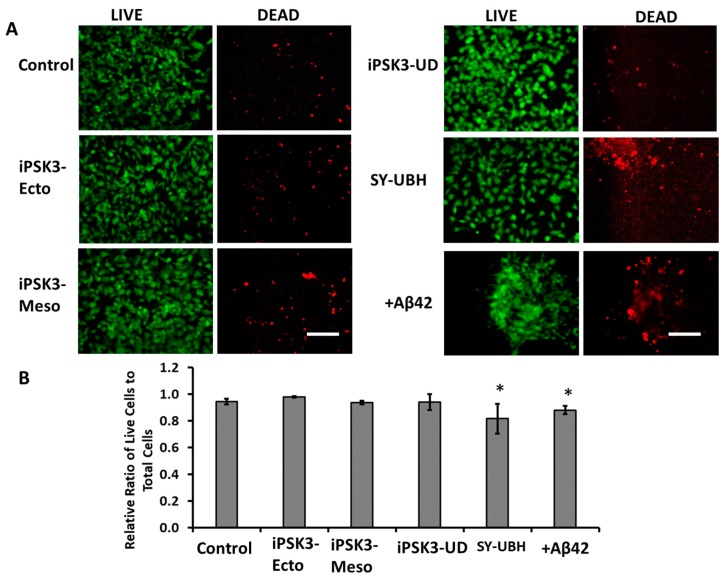
Effects of hiPSC-derived EVs on viability of Aβ42-treated iNPC spheroid outgrowth. Cell viability was examined by LIVE/DEAD assay that utilizes Calcein AM (green) (for live cells) and ethidium homodimer-1 (red) (for dead cells). Day 30 cortical spheroid outgrowth was treated with Aβ42 oligomers and different groups of hiPSC-EVs for three days before the analysis. (**A**) Fluorescent images of LIVE and DEAD cells; Blue: Hoechst. (**B**) Quantification of cell viability (*n* = 3). Scale bar: 100 µm. * indicates *p* < 0.05 compared to the control condition.

**Figure 7 cells-08-00993-f007:**
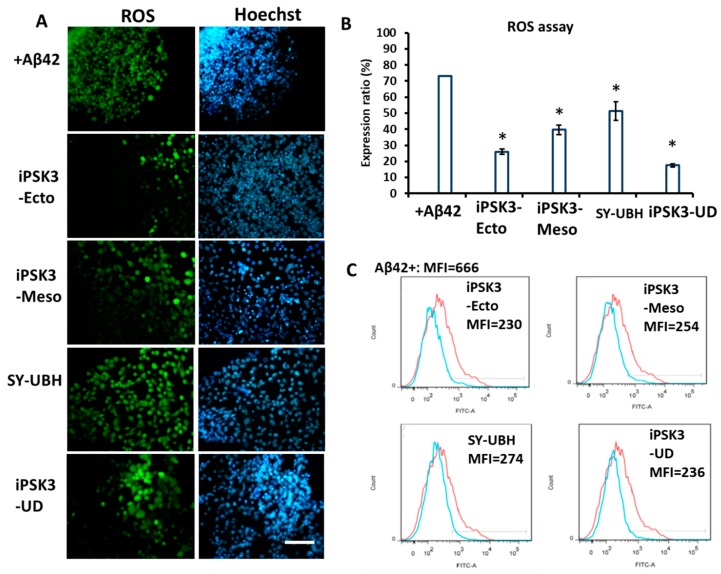
Effects of hiPSC-derived EVs on oxidative stress of Aβ42-treated iNPC spheroid outgrowth. Day 30 cortical spheroid outgrowth was treated with Aβ42 oligomers and different groups of hiPSC-EVs for three days before the analysis. (**A**) Fluorescent images of reactive oxygen species (ROS); Blue: Hoechst. Scale bar: 100 µm. (**B**) Quantification of relative ROS expression (*n* = 3). The expression ratio indicates the ratio of ROS intensity (green) to the total nuclear intensity (blue). The error for the +Aβ42 group is very small/zero. * indicates *p* < 0.05 compared to the culture treated with Aβ42 oligomers. (**C**) Flow cytometry analysis for ROS expression. Red line: iNPCs treated with Aβ42 oligomers (positive control); Blue line: iNPCs treated with Aβ42 oligomers plus a different type of hiPSC-EVs. MFI: mean fluorescence intensity.

**Figure 8 cells-08-00993-f008:**
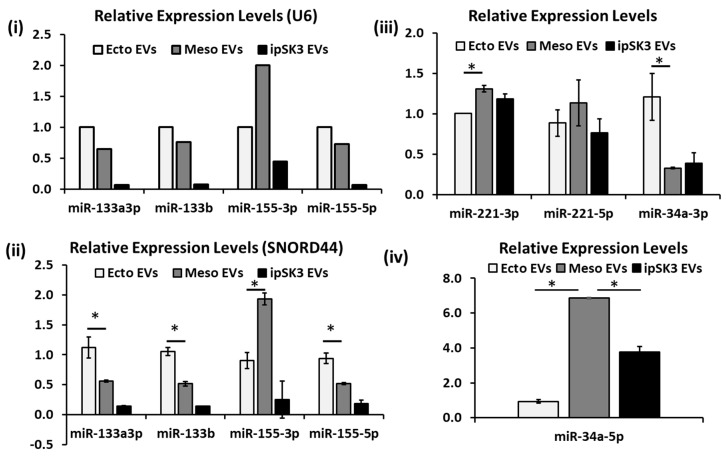
Differential expression of miRNAs in different hiPSC-EV groups. RT-PCR analysis of miR-133 and miR-155 in different EV groups. (**i**) Normalized to U6; (**ii**) normalized to SNORD44 (*n* = 3). (**iii**) RT-PCR analysis of miR-34a-3p and miR-221 in different EV groups (normalized to SNORD44, *n* = 3). (**iv**) RT-PCR analysis of miR-34a-5p in different EV groups (normalized to SNORD44, *n* = 3). * indicates *p* < 0.05.

**Table 1 cells-08-00993-t001:** Primer sequences of different microRNAs.

miR Type	Primer Sequence
hsa-miR-133a-3p	CTTTGGTCCCCTTCAACCAG
hsa-miR-133b	GTTTGGTCCCCTTCAACCA
hsa-miR-155-3p	GCTCCTACATATTAGCATTAACAAAAA
hsa-miR-155-5p	TGCTAATCGTGATAGGGGTAAA
hsa-miR-221-3p	GCGAGCTACATTGTCTGCTG
hsa-miR-221-5p	GCACCTGGCATACAATGTAGA
hsa-miR-34a-3p	ATCAGCAAGTATACTGCCCTAAAA
hsa-miR-34a-5p	GGCAGTGTCTTAGCTGGTTGTAAAA
RNU6	GCAAATTCGTGAAGCGTTCC
SNORD48	CTCTGAGTGTGTCGCTGATGC
SNORD44	AACTGTGTGCTGATTGTCACG

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
