# Peer review of "Differential Effects of Extracellular Vesicles of Lineage-Specific Human Pluripotent Stem Cells on the Cellular Behaviors of Isogenic Cortical Spheroids"

_cells, 2019, doi:10.3390/cells8090993_

Round 1

Reviewer 1 Report

Summary of their study

This study examines the differential expression of exosomes in iPSC-derived lineages and how extracted exosomes could be beneficial for neural cell support both in healthy neural cells as well as Alzheimer’s disease iPSC-derived neural cells. The study provides a nice characterization of EVs produced by iPSC-derived lineages, that is currently under-documented in the literature. No significant difference between the cells genetic background or the general production of EVs from different lineages was found, but the neural specific EV’s showed the ability to reduce oxidative stress of cortical spheroids and support healthier, longer axons.The study further identifies microRNAs associated with the EVs of different lineages.

Strength, weaknesses and recommendation

The major strength of this study is the characterization of EVs in iPSC-derived lineages. The biggest weakness may be that the study is descriptive in nature and does not provide mechanistic insight into the findings as well as the fact that the effects are relatively small. Nevertheless, the study is very well written, the experiments are executed thoroughly, and the literature is discussed adequately. Thus, it is well suited for publication in Cells and we recommend publication with minor revisions.

Major comments:

- figure 3A ii b, the SY-UBH cells seem to produce exosomes (from EM data), but do not express Alix, TSG101, little CD63 and Hsc70, what machinery do the authors speculate that these cells exosomes use then?

- the effect sizes of treatment with exosomes are small (figure 5 and 6). From how many independent differentiations is this data retrieved? Possibly the effect sizes would become more pronounced if more biological repeats would be added, i.e. the error/variability may be reduced.

-it would be interesting to know how treatment of cortical spheroids with exosomes derived from iPSC lineages compared to treatment of the spheroids with exosomes derived from primary sources. Since this is not easily done experimentally, maybe the authors could discuss this based on literature.

Minor comments/suggestions:

-line 142, what does oC stand for (in 2-8oC)?

- Figure 1A i and ii, is it correct that the iPSK3 do not form colonies in the undifferentiated state? But the SY-UBH do form colonies? If this is not correct and they all do form colonies, a picture representing this adequately may be better suited (in i).

- Figure 1B, what is the difference between the two pictures (each in i and ii)?

-line 279 : ‘The differentiation efficiency, as measured by α-actinin, ranged from 23.7% to 84.8% ‘. Is it normal to have such a wide range of efficiency? If so, it may be good to say that also in addition to the reference.

-line 283, suppl. Fig S1b, seem not to be the right reference here.

-Line 284: ‘The neuronal cells contain both glutamatergic and GABAergic neurons’. This statement is not supported with data.

-line 297: what is NTA analysis, it is not spelled out in the text, only in the figure legend

- Figure 2A (a) the iPSK3-ectoderm has no error bar

-line 301: what is mode size, explanation would help?

- Figure 3A, ii b is not well discussed and the labeling is confusing

-Figure 4, what is the control? Is it treated without exosomes, just media?

- Why isn’t SY-UBH included in Fig5?

-Figure 7b, first bar is missing error bar

-the labeling of subfigures A, a, ii is a bit confusing and not very consistant across figures, maybe one could find a common theme and keep it throughout the figures.

-Suppl. Fig 7 is not discussed.

-it would make it easier to reach broader audiences if relevance of specific microRNA’s and other small molecules mentioned were tied in with the introduction, answering the question- why are these important to this particular study?

-Line 53 typo: change “regulate of immune responses”

-in Materials and Methods:

Line 105: confusing sentence Section 2.1: Why did they treat the healthy control cells differently than the disease line prior to differentiation Line 132: they use the abbreviation EB without writing out the full term. In section 2.4 they use rcf and rpm, they should only use one or the other to be consistent In supplementary table S1, I am not sure why they have 2 dilutions for some of the ABs Line 217: they use 96 wp without spelling out “well plates” Line 220. I believe trated should be treated

-figures 4,6,7 it would be easier to read the figures if in the pictures and bar graph the order of the conditions is the same.

Reviewer 2 Report

Differential Effects of Extracellular Vesicles of Lineage-specific Human Pluripotent Stem Cells on Cellular Behaviors of Isogenic Cortical Spheroids

Overview:  Marzano et. al. characterized EVs isolated from iPSC-derived neural progenitors, cardiac cells and undifferentiated iPSCs from “healthy” and Alzheimer associated line.  The authors utilized nanoparticle trafficking, immunoblotting, microRNA analysis, and investigated the neural protective abilities.  The characterization of EVs from different populations may be impactful for future clinical applications of EVs.  The manuscript contained several interesting pieces of data but was presented in a very disconnected manner and did not flow.  For instance the justification of miRNA profiling did not seem to fit in the manuscript.  Several of the figures utilized different labeling techniques and was presented in a very cut/paste manner.    I have several concerns/specific comments below.

Major:

The manuscript would be significantly improved if multiple iPSC lines were utilized…or at the very least a differentiation of SY-UBH to cardiac/neural EVs. The introduction is well written but seems to discuss far more than what is relevant for this paper. I would like to see a more concise introduction focusing on the relevant portions to this manuscript.  Along those same lines a better introduction to Alzheimer’s disease and the neuroprotective role of EVs is needed. Following differentiation into cardiac/neural lineages conditioned medium was collected days 15-20. Did the authors ensure that the proper markers were expressed at day 15?  Immuno (Figure 1) shows day 20.  Additionally what was the purity of the mesoderm/ectoderm differentiations? Unclear if conditioning medium was collected from all days and then pooled and EVs were collected…please clarify. Throughout the manuscript results section several sentences/interpretations of the data would have been more suited to appear in the discussion. Inconsistent labeling/order of labeling of the figures. For example in Figure 2 Aa and 2Ab the order of graphs are not consistent.  Additionally, iPSK3-Mesoderm is labeled as iPSK3-Meso, Meso, MESO-Exo, iPSK3-meso…please be consistent. Double check all figures/legends to ensure that error bars are present and that statistic symbols are clearly represented. Ie Figure 2Bb is # vs. Mode size of the respective group??? was this comparison done by T-test or were all groups compared with ANOVA? Why the two CD63 western blots? Which band is CD63?  Why does the band intensity pattern in Figure 3Aiib not match the pattern in Aiia?  It seems a housekeeping control is missing. Figure 4 images would be greatly improved by providing an overlay image. Figure 7B is a bit confusing…please clarify what relative expression you are referring to. I believe it is in comparison to each experimental group without AB42…so why the missing error bar?  Please add the positive control to the figure as well.

Minor

Please consistently use α-actinin throughout the text/figures. Figure 1. Please use one scale bar color and adjust the size of the scale bar if possible. Red, orange, yellow are extremely difficult to see in the Nanosight figures, please consider using a different color. Figure labeling.. Figure 2 is extremely confusing…labeling Figures as 1A,B,C,D…or Figure 1Ai, ii, iii would be better than Figure 1Aa Please ensure that BrdU method is described in more detail in methods. Figure 5 has multiple images of the same groups…are those duplicates?

Round 2

Reviewer 2 Report

The manuscript was dramatically improved following the first round of revisions.  I have a few minor comments.

1A)  The stats are confusing in Figure 2Aiii and 2Bii.  First, in Figure 2Bii, the legend indicates that *,**,# indicate p<0.05 between the compared conditions and the undifferentiated iPSK3 group and in the response to reviewers # is for the mean size comparison between SY-UUBH group and iPSK3 group.  So why is there a # symbol above both iPSK3 and SY-UBH? Wouldn't it be appropriate to just have a # symbol above the SY-UBH to indicate that there is a difference between SY-UBH and iPSK3?  

1B) In Fig 2Aiii I'm confused why there is a ** symbol above the iPSK3-UD mean and # above the iPSK3-UD mode?  In this figure each bar graph (minus the iPSK3 Ectoderm  mode bar) has a statisitic symbol.

1C) What is the difference between *, **, and #?  If there is no difference just use one symbol.  If there are differences separate them in the legend (ie *p<0.05 vs. iPSK3UD and #p<0.05 vs. mode for each respective experimental group)

2) Simplify figure legend 7 statisitc line to...(ie... *p<0.05 vs AB42 treatment)   

3) Figure 8i is missing error bars...was this an n=1?  
